# Prevalence of Upper Limb Musculoskeletal Disorders and Their Association with Smartphone Addiction and Smartphone Usage among University Students in the Kingdom of Saudi Arabia during the COVID-19 Pandemic—A Cross-Sectional Study

**DOI:** 10.3390/healthcare10122373

**Published:** 2022-11-25

**Authors:** Mohamed Sherif Sirajudeen, Msaad Alzhrani, Ahmad Alanazi, Mazen Alqahtani, Mohamed Waly, Md. Dilshad Manzar, Fatma A. Hegazy, Muhammad Noh Zulfikri Mohd Jamali, Ravi Shankar Reddy, Venkata Nagaraj Kakaraparthi, Radhakrishnan Unnikrishnan, Hariraja Muthusamy, Wafa Alrubaia, Nidaa Alanazi, Faizan Zaffar Kashoo, Mohammad Miraj

**Affiliations:** 1Department of Physical Therapy and Health Rehabilitation, College of Applied Medical Sciences, Majmaah University, Majmaah 11952, Saudi Arabia; 2Department of Medical Equipment Technology, College of Applied Medical Sciences, Majmaah University, Majmaah 11952, Saudi Arabia; 3Department of Nursing, College of Applied Medical Sciences, Majmaah University, Majmaah 11952, Saudi Arabia; 4Department of Physiotherapy, College of Health Sciences, University of Sharjah, Sharjah 27272, United Arab Emirates; 5Faculty of Physical Therapy, Cairo University, Cairo 11829, Egypt; 6Department of Physiotherapy, Universiti Tunku Abdul Rahman, Kajang 43000, Malaysia; 7Department of Medical Rehabilitation Sciences, College of Applied Medical Sciences, King Khalid Univesity, Abha 61413, Saudi Arabia

**Keywords:** musculoskeletal pain, smartphone overuse, physical activity, ergonomics

## Abstract

This study aimed to investigate the prevalence of upper limb musculoskeletal disorders (MSDs) and their association with smartphone addiction and smartphone usage among university students in the Kingdom of Saudi Arabia during the COVID-19 pandemic. A total of 313 university students aged 18 years and older who owned a smartphone and used it during the preceding 12 months participated in this cross-sectional study. The prevalence of upper limb MSDs, smartphone addiction/overuse, and levels of physical activity were recorded using the standardized Nordic musculoskeletal questionnaire, the smartphone addiction scale (short version), and the international physical activity questionnaire (short form), respectively. Data collection was performed on campus between March and May 2021. Binary logistic regression was used to determine the association between the prevalence of upper limb MSDs and smartphone addiction/overuse and levels of physical activity. The 12-month prevalence of MSDs of the shoulder, elbow, and wrist/hand regions due to smartphone use among participants was found to be 20.13%, 5.11%, and 13.42%, respectively. Shoulder (odds ratio (OR) = 11.39, 95% confidence interval (CI) = 4.64–27.94, *p* < 0.001), elbow (OR = 15.38, 95% CI = 1.92–123.26, *p* = 0.01), and wrist/hand MSDs (OR = 7.65, 95% CI = 2.75–21.22, *p* < 0.001) were more prevalent among participants who were categorized as having smartphone addiction/overuse measures. Promoting awareness about the healthy use of smartphones, including postural education and decreasing screen time, is necessary to reduce smartphone-related MSDs.

## 1. Introduction

In this era of digital and advanced connectivity, there has been an exponential increase in smartphone usage across the globe for internet access, social networking, educational purposes, gaming, and other daily life activities [1]. The smartphone has become an imperative gadget in modern society, and the extended usage of smartphones leads to addiction and other physical problems for users [2,3,4]. Smartphone addiction has emerged as a crucial global concern in recent times, affecting people at all levels, including students. In the Kingdom of Saudi Arabia, there has been a steep rise in the prevalence of smartphone addiction among university students, from 19.1% to 60.3% reported in 2016 and 2019, respectively [5,6]. This prevalence increased further during COVID-19 pandemic lockdowns and social isolation [5], whereby online platforms were heavily used for educational and social needs [6]. Addiction symptoms include a longing for one’s smartphone, withdrawal, tolerance, disturbances in daily life, and an inclination towards virtual online community companionship [7,8]. In such a setting, addiction and subsequent overuse are associated with memory and/or attention problems, resulting in a significant reduction in academic performance and health-related quality of life among students [5,9,10]. Additionally, researchers have also reported an association between smartphone addiction and eating disorders, insomnia, an increased risk of road traffic accidents (using smartphones while driving), and musculoskeletal symptoms [11,12,13,14].

Smartphone addiction and overuse are associated with musculoskeletal disorders (MSDs), mostly affecting the neck (55.8% to 89.9%), followed by the shoulders (37.8% to 71.6%), hands/wrists (13% to 32%), and elbows (14.1% to 15%), among university students and general users [15]. This could be attributed to poor posture for an extended period, i.e., flexed neck and unsupported elbows, coupled with repetitive movements of the thumb to scroll through the screen, leading to undue static loads on the neuromusculoskeletal structures in the cervical and upper extremity regions while using a smartphone [15,16,17,18]. Most of the earlier research that aimed to study the association between smartphone addiction and upper extremity disorders among university students in Saudi Arabia suffers from methodological issues and limitations [5,6]. This included not considering the contribution of smartphone use to musculoskeletal symptoms and lacking a standardized definition of musculoskeletal disorder encompassing parameters such as the intensity, duration, and frequency of neck symptoms, which is crucial for discriminating between major and minor MSDs. Therefore, this study aimed to investigate the prevalence of upper limb musculoskeletal disorders and their association with smartphone addiction and usage among university students in the Kingdom of Saudi Arabia during the COVID-19 pandemic. The findings of this study are crucial for providing input in initiating awareness and better design strategies to prevent the occurrence of upper extremity disorders among smartphone users.

## 2. Materials and Methods

### 2.1. Participants

A total of 380 students aged 18–45 years from the College of Applied Medical Sciences, Majmaah University, Kingdom of Saudi Arabia, were invited to participate in this cross-sectional study. Data collection was performed on campus between March and May 2021. Individuals who owned and used a smartphone in the preceding 12 months were included. Those with a history of cervical fractures or surgeries, congenital or acquired musculoskeletal deformities, or neurological diseases or who were pregnant were excluded [19,20,21].

### 2.2. Ethics Statement

The ethical guidelines recommended in the Declaration of Helsinki (1964) were followed in all stages of the study. The Majmaah University Research Ethics Committee issued ethical approval for this study (MUREC-Dec.30/COM-2020/18-2). All participants gave written informed consent in English before their enrollment in the study. The participants’ privacy and anonymity were protected, and no identifying information was obtained through the study questionnaire. This study did not include any minor participants.

### 2.3. Measurement

The questionnaire used in this study consisted of six sections relating to socio-demographics, smartphone usage, the standardized Nordic musculoskeletal questionnaire (upper limb regions), smartphone addiction, and physical activity. The socio-demographic section comprised items relating to age, gender, height, weight, hand dominance, department, and level of education. Body mass index (BMI) was determined by dividing participants’ weight (kilograms) by their height (meters squared) [22]. The smartphone usage section consisted of items such as the duration of smartphone, tablet, and laptop computer usage in years, the duration of daily smartphone use in general and for specific purposes, such as study activities, social media, and playing games, and patterns adopted when holding the smartphone.

#### 2.3.1. Standardized Nordic Musculoskeletal Questionnaire

A component of the standardized Nordic musculoskeletal questionnaire (SNMQ) addressing the shoulder, elbow, wrist, and hand regions was utilized to determine musculoskeletal disorders in the neck region. SNMQ is a valid and reliable instrument and is widely used in epidemiological studies to screen musculoskeletal symptoms [23]. Participants recorded symptoms such as pain, numbness, tingling, aching, stiffness, and burning in the upper extremity region that they had experienced during or after smartphone use in the preceding 12 months. The participants also reported the intensity/severity, duration, and frequency of their upper extremity symptoms. A musculoskeletal disorder was defined by the experience of the symptoms listed above with moderate pain or more that lasted for a minimum of one week or occurred at least once a month during the preceding 12 months [24]. Researchers affiliated with the National Institute for Occupational Safety and Health developed and standardized this case definition [25].

#### 2.3.2. Smartphone Addiction Scale (Short Version) (SAS-SV)

The SAS-SV was used to determine smartphone addiction/overuse. The SAS-SV comprises 10 self-report items and is scored on a 6-point Likert scale, where “1” represents “strongly disagree”, and “6” denotes “strongly agree”. The overall score of the SAS-SV ranges from 10 to 60 and is directly proportional to the extent of smartphone use in the past year. The psychometric properties of the SAS-SV, such as content and criterion validity and internal consistency (Cronbach’s alpha: 0.91), were found to be adequate. Scores of ≥ 31 and ≥ 33 denote smartphone addiction/overuse among males and females, respectively [8]. Researchers have also previously employed this cut-off to screen smartphone addiction/overuse among university students [6,26].

#### 2.3.3. International Physical Activity Questionnaire (Short Form) (IPAQ-SF)

We used the IPAQ-SF to assess the physical activity of study participants. The IPAQ-SF is a valid and reliable self-report questionnaire consisting of 9 items used to recall physical activities accomplished during the preceding 7 days. Data collected using the IPAQ-SF are used to determine participants’ metabolic equivalents of task (METs) and are categorized as light-intensity (less than 3 METs), moderate-intensity (3 to 6 METs), and vigorous-intensity activities (more than 6 METs) [27,28].

### 2.4. Pretesting

A five-member expert panel consisting of two physical therapists, two orthopedic surgeons, and one public health physician evaluated the comprehensibility of the questionnaire. A sample of 30 university students participated in the pretesting. Members of the expert panel and participants of the pretesting agreed that the questionnaire was clear and easy to understand for university students.

### 2.5. Sample Size Calculation

The sample size was determined using the sample size calculation for estimating a single proportion. We used the prevalence of smartphone addiction of 60.3% reported among Qassim University’s students [6]. The required sample size was identified as 277 with 95% confidence and 5% absolute precision.

### 2.6. Statistical Analysis

Data were analyzed using SPSS (version 23.0, IBM Corp, Armonk, NY, USA) for Windows. Descriptive statistics were produced for socio-demographic characteristics, smartphone usage, and the prevalence of musculoskeletal disorders of the upper limb. The prevalence of MSDs in the shoulder, elbow, and wrist/hand regions was determined by dividing the number of participants categorized as having MSDs in the respective regions based on the case definition by the total number of study participants. A binary logistic regression analysis (Wald Chi-squared test) was used to determine the association between the study variables and the presence/absence of MSDs in the participants’ shoulder, elbow, and wrist/hand regions. The selection of predictors in binary logistic regression models was based on a *p* value (0.20) for the bivariate association between dependent and independent variables. However, weight, BMI, the duration of laptop use (in years), the duration of smartphone use (in years), hand dominance, and physical activity were used as predictors based on theoretical justification, as previous studies have reported their association with MSDs [29,30,31,32,33] even though they did not satisfy the statistical consideration. The statistical significance was set at a 5% probability level.

## 3. Results

A total of 313 students among the 380 invited students participated in this study. The participation rate was 82.4%. The socio-demographic characteristics of the participants are presented in Table 1. The mean age of the participants was 22.6 (4.08) years. Most of the participants were female students (54.3%). The mean BMI of the participants was 23.92 (5.13) Kg/m^2^. Most of the participants were right-hand-dominant (88.2%). More than half of the participants were physical therapy students (53.3%). Most of the participants were bachelor-level students (84.7%). Regarding the self-reporting of physical activity, most participants were categorized as engaging in light physical activity (45.7%), whereas 39.3% and 15% belonged to moderate and vigorous physical activity categories, respectively.

Study participants reported a mean duration of 9.58 years of smartphone use, 3.97 years of tablet use, and 6.95 years of laptop use. Most participants (52.1%) reported using their smartphones for 7 h or more daily. A total of 119 participants (38%) used their smartphones for less than an hour daily for study purposes. However, 44.7% of participants spent 4 h or more daily on social media platforms using their smartphones. Most participants spent less than an hour daily playing games on their smartphones. About 55% of participants reportedly used their right hand to hold their smartphones. The prevalence of smartphone overuse/addiction among participants was 55.3% (Table 2).

The 12-month prevalence of MSDs of the shoulder, elbow, and wrist/hand regions due to smartphone use among the participants was found to be 20.13%, 5.11%, and 13.42%, respectively. The results of the binary logistic regression analysis regarding the association between socio-demographic characteristics and the prevalence of shoulder MSDs are presented in Table 3. Increased age (odds ratio (OR) = 1.13, 95% confidence interval (CI) = 1.05–1.22, *p* = 0.001) and female gender (OR = 4.89, 95% CI = 1.84–12.99, *p* = 0.001) were associated with the occurrence of shoulder MSDs. The results of the binary logistic regression analysis regarding the association between smartphone usage and the prevalence of shoulder MSDs are presented in Table 4. Shoulder MSDs (OR = 11.39, 95% CI = 4.64–27.94, *p* < 0.001) were more prevalent among participants who were categorized as having smartphone addiction/overuse measures. The results of the binary logistic regression analysis regarding the association between socio-demographic characteristics and the prevalence of elbow MSDs are presented in Table 5. Decreased participant height (OR = 0.93, 95% CI = 0.87–0.99, *p* = 0.032) was associated with the occurrence of elbow MSDs. The results of the binary logistic regression analysis regarding the association between smartphone usage and the prevalence of elbow MSDs are presented in Table 6. Elbow MSDs (OR = 15.38, 95% CI = 1.92–123.26, *p* = 0.01) were more prevalent among participants who were categorized as having smartphone addiction/overuse measures. The results of the binary logistic regression analysis regarding the association between socio-demographic characteristics and the prevalence of wrist/hand MSDs are presented in Table 7. Female gender (OR = 6.36, 95% CI = 1.98–20.45, *p* = 0.002) and increased participant weight (OR = 1.04, 95% CI = 1–1.07, *p* = 0.044) were associated with the occurrence of wrist/hand MSDs. The results of the binary logistic regression analysis regarding the association between smartphone usage and the prevalence of wrist/hand MSDs are presented in Table 8. Increased duration of tablet usage (OR = 1.1, 95% CI = 1–1.2, *p* = 0.037) and smartphone addiction/overuse (OR = 7.65, 95% CI = 2.75–21.22, *p* < 0.001) were significantly associated with the occurrence of wrist/hand MSDs (OR = 1.1, 95% CI = 1–1.2, *p* = 0.037). Wrist/hand MSDs were more prevalent among participants who reported holding their smartphone in their left hand (OR = 7.66, 95% CI = 1.85–31.75, *p* = 0.005) compared to participants holding their smartphone in their right hand or both hands.

## 4. Discussion

This study determined the prevalence of smartphone addiction, smartphone usage, and their association with the prevalence of MSDs in the upper limbs. It also recorded socio-demographic data such as age, gender, hand dominance, physical activity, etc. The distribution of hand dominance among participants in this study (right, left, and ambidextrous) was identical to participants in similar studies from Australia [34] and Thailand [21]. Moreover, a recent meta-analysis by Papadatou-Pastou et al. also reported on hand dominance distribution, which was consistent with that reported in our study [35]. In relation to the physical activity of the participants, the majority of them were categorized as engaging in light physical activity, followed by moderate physical activity, and a few engaged in vigorous physical activity. In comparison with earlier research reporting on the physical activities of university students during the COVID-19 pandemic using the IPAQ-SF, the physical activity of our participants was similar to Turkish students [36], less compared to American students [37], and better than that of Jordanian students [38]. Earlier studies reported a sharp decline in physical activity among university students due to COVID-19 lockdowns and related restrictions [37,39,40,41,42].

This study found that 55.3% of participants reported smartphone addiction. This finding is comparable to an earlier report in 2019 among university students in Saudi Arabia (60.3%) before lockdowns. In other words, university students in Saudi Arabia already experienced smartphone addiction before lockdowns and continued to experience it after. As the Kingdom of Saudi Arabia imposed measures, including partial nationwide lockdowns, social distancing, and the closure of educational institutions to prevent and control the COVID-19 pandemic [43], educational institutions utilized online-based learning platforms to teach and assess students’ academic performance [44]. Additionally, due to social isolation, students tend to use smartphones to virtually connect to the online community through social networking sites and spend considerable time playing phone-based games, browsing internet sites, and watching social media. This increases the time spent on smartphones, which could lead to negative consequences, such as smartphone addiction [45]. Smartphone addiction/overuse was associated with a decline in academic performance, musculoskeletal pain, poor sleep, stress, anxiety, and negative emotions among university students [46]. A study by Hosen et al. reported an alarming level (86.9%) of smartphone addiction among Bangladesh students during the COVID-19 pandemic [44]. The prevalence of smartphone overuse among participants in this study (55.3%) is slightly less compared to the rates reported among university students in the Jeddah (63%) and Makkah regions (67%) in Saudi Arabia during the COVID-19 pandemic [46,47].

The 12-month prevalence of MSDs of the shoulder, elbow, and wrist/hand regions due to smartphone use among participants was found to be 20.13%, 5.11%, and 13.42%, respectively. The prevalence rates reported in this study are lower compared to those reported in a recent systematic review by Zirek et al. (pain in the shoulder: 37.8% to 71.6%; elbow: 14.1% to 15%; and wrist/hand: 13% to 32%) [15]. This ambiguity among earlier researchers in reporting the prevalence of musculoskeletal symptoms may be attributed to methodological differences. It is recommended to screen for the occurrence of musculoskeletal disorders based on parameters such as the intensity, duration, and frequency of the presenting symptom at the anatomical location to determine significant cases and exclude minor ones, which is a crucial element in reporting epidemiological studies [48]. Most of the earlier studies among similar populations did not clearly state whether they recorded details of the intensity, duration, and frequency of the presenting symptom [6,49].

The findings of this study revealed that musculoskeletal symptoms in the shoulder, elbow, and wrist/hand region were associated with smartphone addiction/overuse and are consistent with earlier research among similar populations [33,50,51]. Individuals who were categorized as experiencing smartphone overuse/addiction adopt faulty postures, as their necks are flexed and their elbows are unsupported during extensive use of their smartphones, resulting in profuse static load in the neck and upper extremities [16]. A systematic review by Eitivipart et al. also reported augmented activity in the shoulder and forearm muscles, which may be attributed to the onset of muscle fatigue and a pain threshold decline during smartphone use [52]. Moreover, users hold their smartphones in one hand and scroll using one finger. These monotonous activities of the wrist and thumb subject the joints to excessive wear and tear; raise the pressure in the carpel tunnel region, compress its contents, and cause swelling of the median nerve and flexor pollicis longus tendon; and result in musculoskeletal symptoms [17,18,52]. Excessive smartphone use was also reported to be associated with an impaired ability to perform fine movements using the thumb [53].

Shoulder and wrist/hand symptoms were found to be more prevalent among female participants compared to their male counterparts, which is similar to previous studies [54,55] and may be attributed to underlying gender-related psychological and biological factors. Females exhibit a greater tendency to recognize and report symptoms compared to males. There are also gender differences regarding musculoskeletal architecture, metabolic functions, and hormonal influences inducing pain-related parameters, such as perception and threshold [48,56]. In this study, the prevalence of wrist/hand MSDs was higher among participants who reported holding their smartphone in their left hand (31.6%) compared to those using both hands (10.2%). Using a single hand to hold the smartphone while scrolling increases repetitive movements and muscle activity compared to using both hands, which increases the risk of developing MSDs [57,58].

Previous studies have proposed some crucial recommendations to reduce the risk of developing MSDs among smartphone users, which are worth discussing here. Thorburn et al. reported that musculoskeletal symptoms were less prevalent among study participants who used a cradle or any other external support to hold their smartphones and those who frequently changed their postures (once every 5 min) during smartphone use [34]. Only 1.3% of participants in this study reported using a cradle or any other external support to hold their smartphones. Smartphone users in an earlier study reported the onset of musculoskeletal symptoms typically after 15 to 30 min of device usage. Taking into consideration this threshold time for musculoskeletal symptoms among smartphone users, the onset of symptoms could be prevented to a greater extent if users restricted device usage to 15 min per session [34].

In this study, the prevalence of upper limb MSDs was not associated with the level of physical activity. A systematic review by Mansi et al. reported the efficacy of physical activity in the prevention of hip fractures and the reduction in neck, shoulder, and lower back pain [59]. Regular physical activity was beneficial for improving bone mineral density and muscle capillary density, which could partially support physical activity’s role in reducing musculoskeletal disorders [60,61,62].

### Limitations

The cross-sectional nature of the research design exercised in this study could not confirm a causal relationship between associated variables and MSDs in the upper limbs. Data obtained from participants were self-reported and inherited the risk of recall bias. The convenience sampling technique employed in this study limits us from generalizing the findings to Saudi Arabia as a whole. Some of the variables had wide confidence intervals, which suggests that the odds ratios for such variables had low precision [63].

## 5. Conclusions

The study findings show that one in every five smartphone users reported musculoskeletal symptoms in the shoulder region. Smartphone addiction was significantly associated with the occurrence of MSDs in the upper limb. Measures to promote awareness of the healthy use of smartphones, including postural education and decreasing screen time, are warranted.

## Figures and Tables

**Table 1 healthcare-10-02373-t001:** Socio-demographic characteristics.

Characteristics	Mean (SD)/Frequency (%)
Age (Years)	22.6 (±4.08)
Gender	
*Male*	143 (45.7%)
*Female*	170 (54.3%)
Height (cm)	164.42 (±10.16)
Weight (kg)	65.97 (±18.54)
Body mass index (Kg/m^2^)	23.92 (±5.13)
Hand dominance	
*Right*	276 (88.20%)
*Left*	31 (9.91%)
*Both equally used (ambidextrous)*	6 (1.91%)
Department	
*Physical therapy*	167 (53.35%)
*Nursing*	51 (16.29%)
*Medical laboratory sciences*	29 (9.26%)
*Medical equipment technology*	33 (10.54%)
*Medical imaging*	33 (10.54%)
Education level	
*Bachelor*	265 (84.66%)
*Postgraduate/Master*	48 (15.34%)
Physical activity	
*Light*	143 (45.70%)
*Moderate*	123 (39.29%)
*Vigorous*	47 (15.01%)

SD—Standard deviation.

**Table 2 healthcare-10-02373-t002:** Smartphone usage of the participants.

Characteristics	Mean (SD)/Frequency (%)
Smartphone and other gadget usage (years)	
*Smartphone*	9.58 (±2.66)
*Tablet*	3.97 (±3.84)
*Laptop*	6.95 (±4.74)
Daily smartphone use	
*About an hour*	2 (0.6%)
*1–3 h*	22 (7%)
*3–5 h*	51 (16.3%)
*5–7 h*	75 (24%)
*7 h or more*	163 (52.1%)
*Purpose of smartphone use*	
Study	
*Less than an hour*	119 (38%)
*1–2 h*	53 (16.9%)
*2–3 h*	42 (13.4%)
*3–4 h*	38 (12.1%)
*4 h or more*	61 (19.5%)
Social media	
*Less than an hour*	16 (5.1%)
*1–2 h*	36 (11.5%)
*2–3 h*	45 (14.4%)
*3–4 h*	76 (24.3%)
*4 h or more*	140 (44.7%)
Playing games	
*Less than an hour*	209 (66.7%)
*1–2 h*	36 (11.5%)
*2–3 h*	18 (5.8%)
*3–4 h*	18 (5.8%)
*4 h or more*	32 (10.2%)
Holding the smartphone	
*Right hand*	172 (55%)
*Left hand*	19 (6.1%)
*Both hands*	118 (37.7%)
*Use of cradle, stand, table or other rest*	4 (1.3%)
Smartphone addiction	
*Overuse*	173 (55.3%)
*Non-overuse*	140 (44.7%)

SD—Standard deviation.

**Table 3 healthcare-10-02373-t003:** Association between shoulder MSDs and participant characteristics.

Characteristics	Shoulder MSDs		Significance	Adjusted OR (95% CI)
	Yes	No	*p* Value ^#^	
	Mean (SD)/Frequency (%)	Mean (SD)/Frequency (%)		
Age (Years)	23.3 (±5.07)	22.46 (±3.79)	0.001 *	1.129 (1.048–1.217) *
Gender				
*Male*	16 (11.2%)	127 (88.8%)	-ref	-ref
*Female*	47 (27.6%)	123 (72.4%)	0.001 *	4.895 (1.845–12.986) *
Height (cm)	160.62 (±13.52)	165.39 (±8.9)	0.356	0.981 (0.941–1.022)
Weight (kg)	66.06 (±20.08)	65.96 (±18.18)	0.404	1.015 (0.981–1.049)
Body mass index (Kg/m^2^)	24.21 (±5.17)	23.86 (±5.13)	0.871	0.991 (0.889–1.104)
Hand dominance				
*Right*	56 (20.3%)	220 (79.7%)	-ref	-ref
*Left or both equally used (ambidextrous)*	7 (18.9%)	30 (81.1%)	0.414	1.464 (0.586–3.654)
Physical activity				
*Light*	116 (81.12%)	27 (18.88%)	0.935	0.962 (381–2.429)
*Moderate*	95 (77.24%)	28 (22.76%)	0.697	1.2 (0.479–3.008)
*Vigorous*	39 (82.98%)	8 (17.02%)	-ref	-ref

CI—Confidence interval; OR—odds ratio; SD—standard deviation; *—significant (*p* < 0.05); ^#^—*p* value for Wald Chi-square test. The dependent variable was the dichotomous measure of the presence/absence of shoulder MSDs; the independent variables were age, gender, height, weight, body mass index, hand dominance, and physical activity.

**Table 4 healthcare-10-02373-t004:** Association between shoulder MSDs and smartphone usage.

Characteristics	Shoulder MSDs		Significance	Adjusted OR (95% CI)
	Yes	No	*p* Value ^#^	
	Mean (SD)/Frequency (%)	Mean (SD)/Frequency (%)		
Smartphone and other gadget usage (years)				
*Smartphone*	9.444 (±3.267)	9.62 (±2.491)	0.216	0.924 (0.816–1.047)
*Tablet*	4.746 (±4.348)	3.778 (±3.695)	0.076	1.077 (0.992–1.168)
*Laptop*	7.143 (±5.102)	6.902 (±4.66)	0.869	1.006 (0.939–1.077)
Daily smartphone use				
*Less than 5 h*	8 (10.7%)	67 (89.3%)	-ref	-ref
*5 h or more*	55 (23.1%)	183 (76.9%)	0.053	2.546 (0.99–6.551)
** *Purpose of smartphone use* **				
Study				
*Less than 3 h*	40 (18.7%)	174 (81.3%)	-ref	-ref
*3 h or more*	23 (23.2%)	76 (76.8%)	0.501	1.262 (0.641–2.486)
Social media				
*Less than 3 h*	19 (19.6%)	78 (80.4%)	-ref	-ref
*3 h or more*	44 (20.4%)	172 (79.6%)	0.074	0.503 (0.237–1.068)
Playing games				
*Less than 3 h*	51 (19.4%)	212 (80.6%)	-ref	-ref
*3 h or more*	12 (24%)	38 (76%)	0.675	1.183 (0.538–2.604)
Holding the smartphone				
*Right hand*	35 (20.3%)	137 (79.7%)	0.857	1.061 (0.56–2.009)
*Left hand*	4 (21.1%)	15 (78.9%)	0.923	0.934 (0.231–3.78)
*Both hands/use of cradle, stand, table, or other rest*	24 (19.7%)	98 (80.3%)	-ref	-ref
Smartphone addiction				
*Overuse*	57 (32.9%)	116 (67.1%)	<0.001 *	11.392 (4.644–27.944)
*Non-overuse*	6 (4.3%)	134 (95.7%)	-ref	-ref

CI—Confidence interval; OR—odds ratio; SD—standard deviation; *—significant (*p* < 0.05); ^#^—*p* value for Wald Chi-square test. The dependent variable was the dichotomous measure of the presence/absence of shoulder MSDs; the independent variables were smartphone and other gadget usage (smartphone, tablet, and laptop), daily smartphone use, purpose of smartphone use (study, social media, and playing games), holding the smartphone, and smartphone addiction.

**Table 5 healthcare-10-02373-t005:** Association between elbow MSDs and participant characteristics.

Characteristics	Elbow MSDs		Significance	Adjusted OR (95% CI)
	Yes	No	*p* Value ^#^	
	Mean (SD) Frequency (%)	Mean (SD)/Frequency (%)		
Age (Years)	23.563 (±4.442)	22.576 (±4.065)	0.1	1.108 (0.981–1.253)
Gender				
*Male*	4 (2.8%)	139 (97.2%)	-ref	-ref
*Female*	12 (7.1%)	158 (92.9%)	0.501	1.886 (0.297–11.97)
Height (cm)	154.688 (±22.849)	164.954 (±8.774)	0.032 *	0.93 (0.871–0.994) *
Weight (kg)	70.563 (±27.154)	65.732 (±17.997)	0.883	0.996 (0.942–1.052)
Body mass index (Kg/m^2^)	25.239 (±4.294)	23.855 (±5.172)	0.553	1.054 (0.885–1.255)
Hand dominance				
*Right*	12 (4.4%)	263 (95.6%)	-ref	-ref
*Left or both equally used (ambidextrous)*	4 (10.5%)	34 (89.5%)	0.211	0.450 (0.129–1.572)
Physical activity				
*Light*	8 (5.6%)	135 (94.4%)	0.909	0.916 (0.203–4.127)
*Moderate*	5 (4.1%)	118 (95.9%)	0.501	0.588 (0.125–2.764)
*Vigorous*	3 (6.4%)	44 (93.6%)	-ref	-ref

CI—Confidence interval; OR—odds ratio; SD—standard deviation; *—significant (*p* < 0.05); ^#^—*p* value for Wald Chi-square test. The dependent variable was the dichotomous measure of the presence/absence of elbow MSDs; the independent variables were age, gender, height, weight, body mass index, hand dominance, and physical activity.

**Table 6 healthcare-10-02373-t006:** Association between elbow MSDs and smartphone usage.

Characteristics	Elbow MSDs		Significance	Adjusted OR (95% CI)
	Yes	No	*p*-Value ^#^	
	Mean (SD) Frequency (%)	Mean (SD)/Frequency (%)		
Smartphone and other gadget usage (years)				
*Smartphone*	9.5 (±2.449)	9.589 (±2.675)	0.428	0.911 (0.722–1.148)
*Tablet*	3 (±4.082)	4.025 (±3.834)	0.296	0.92 (0.787–1.075)
*Laptop*	7.563 (±5.138)	6.918 (±4.73)	0.444	1.049 (0.928–1.186)
Daily smartphone use				
*Less than 5 h*	1 (1.3%)	74 (98.7%)	-ref	-ref
*5 h or more*	16 (6.7%)	222 (93.3%)	0.322	3.101 (0.33–29.163)
*Purpose of smartphone use*				
Study				
*Less than 3 h*	7 (3.3%)	207 (96.7%)	-ref	-ref
*3 h or more*	10 (10.1%)	89 (89.9%)	0.108	2.535 (0.814–7.896)
Social media				
*Less than 3 h*	3 (3.1%)	94 (96.9%)	-ref	-ref
*3 h or more*	14 (6.5%)	202 (93.5%)	0.62	1.46 (0.328–6.502)
Playing games				
*Less than 3 h*	12 (4.6%)	251 (95.4%)	-ref	-ref
*3 h or more*	5 (10%)	45 (90%)	0.218	2.157 (0.634–7.335)
Holding the smartphone				
*Right hand*	5 (2.8%)	171 (97.2%)	0.041 *	0.294 (0.091–0.951) *
*Left hand*	2 (10.5%)	17 (89.5%)	0.752	1.369 (0.195–9.617)
*Both hands/use of cradle, stand, table, or other rest*	10 (8.5%)	108 (91.5%)	-ref	-ref
Smartphone addiction				
*Overuse*	15 (8.7%)	158 (91.3%)	0.01 *	15.376 (1.918–123.259) *
*Non-overuse*	1 (0.7%)	139 (99.3%)	-ref	-ref

CI—Confidence interval; OR—odds ratio; SD—standard deviation; *—significant (*p* < 0.05); ^#^—*p* value for Wald Chi-square test. The dependent variable was the dichotomous measure of the presence/absence of elbow MSDs; the independent variables were smartphone and other gadget usage (smartphone, tablet, and laptop), daily smartphone use, purpose of smartphone use (study, social media, and playing games), holding the smartphone, and smartphone addiction.

**Table 7 healthcare-10-02373-t007:** Association between wrist/hand MSDs and participant characteristics.

Characteristics	Wrist/Hand MSDs		Significance	Adjusted OR (95% CI)
	Yes	No	*p* Value ^#^	
	Mean (SD)/Frequency (%)	Mean (SD)/Frequency (%)		
Age (Years)	22.714 (±3.763)	22.613 (±4.137)	0.172	1.065 (0.973–1.166)
Gender				
*Male*	10 (7%)	133 (93%)	-ref	-ref
*Female*	32 (18.8%)	138 (81.2%)	0.002 *	6.359 (1.978–20.448) *
Height (cm)	160.671 (±15.788)	165.012 (±8.885)	0.921	0.998 (0.955–1.042)
Weight (kg)	69.043 (±24.237)	65.504 (±17.507)	0.044 *	1.036 (1.001–1.073) *
Body mass index (Kg/m^2^)	24.643 (±5.941)	23.815 (±5.001)	0.507	0.963 (0.861–1.077)
Hand dominance				
*Right*	34 (12.3%)	242 (87.7%)	-ref	-ref
*Left or both equally used (ambidextrous)*	8 (21.6%)	29 (78.4%)	0.321	0.632 (0.256–1.564)
Physical activity				
*Light*	18 (12.6%)	125 (87.4%)	0.557	0.74 (0.271–2.022)
*Moderate*	17 (13.8%)	106 (86.2%)	0.280	0.657 (0.238–1.815)
*Vigorous*	7 (14.9%)	40 (85.1%)	-ref	-ref

CI—Confidence interval; OR—odds ratio; SD—standard deviation; *—significant (*p* < 0.05); ^#^—*p* value for Wald Chi-square test. The dependent variable was the dichotomous measure of the presence/absence of wrist/hand MSDs; the independent variables were age, gender, height, weight, body mass index, hand dominance, and physical activity.

**Table 8 healthcare-10-02373-t008:** Association between wrist/hand MSDs and smartphone usage.

Characteristics	Wrist/Hand MSDs		Significance	Adjusted OR (95% CI)
	Yes	No	*p*-Value ^#^	
	Mean (SD)/Frequency (%)	Mean (SD)/Frequency (%)		
Smartphone and other gadget usage (years)				
*Smartphone*	10 (±2.802)	9.52 (±2.637)	0.408	1.064 (0.919–1.232)
*Tablet*	5.5 (±4.549)	3.736 (±3.68)	0.037 *	1.105 (1.006–1.213) *
*Laptop*	6.5 (±4.374)	7.02 (±4.804)	0.086	0.931 (0.858–1.01)
Daily smartphone use				
*Less than 5 h*	4 (5.3%)	71 (94.7%)	-ref	-ref
*5 h or more*	38 (16%)	200 (84%)	0.279	1.928 (0.587–6.334)
** *Purpose of smartphone use* **				
Study				
*Less than 3 h*	25 (11.7%)	189 (88.3%)	-ref	-ref
*3 h or more*	17 (17.2%)	82 (82.2%)	0.566	1.26 (0.572–2.776)
Social media				
*Less than 3 h*	8 (8.2%)	89 (91.8%)	-ref	-ref
*3 h or more*	34 (15.7%)	182 (84.3%)	0.440	1.465 (0.555–3.863)
Playing games				
*Less than 3 h*	32 (12.2%)	231 (87.8%)	-ref	-ref
*3 h or more*	10 (20%)	40 (80%)	0.206	1.745 (0.737–4.134)
Holding the smartphone				
*Right hand*	24 (14%)	148 (86%)	0.275	1.55 (0.705–3.405)
*Left hand*	6 (31.6%)	13 (68.4%)	0.005 *	7.661 (1.848–31.754) *
*Both hands/use of cradle, stand, table, or other rest*	12 (9.8%)	110 (90.2%)	-ref	-ref
Smartphone addiction				
*Overuse*	37 (21.4%)	136 (78.6%)	<0.001 *	7.646 (2.754–21.224) *
*Non-overuse*	5 (3.6%)	135 (96.4%)	-ref	-ref

CI—Confidence interval; OR—odds ratio; SD—standard deviation; *—significant (*p* < 0.05); ^#^—*p* value for Wald Chi-square test. The dependent variable was the dichotomous measure of the presence/absence of wrist/hand MSDs; the independent variables were smartphone and other gadget usage (smartphone, tablet, and laptop), daily smartphone use, purpose of smartphone use (study, social media, and playing games), holding the smartphone, and smartphone addiction.

## Data Availability

The data presented in this study are available on request from the corresponding author.

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
