# Peer review of "Prevalence of Upper Limb Musculoskeletal Disorders and Their Association with Smartphone Addiction and Smartphone Usage among University Students in the Kingdom of Saudi Arabia during the COVID-19 Pandemic—A Cross-Sectional Study"

_healthcare, 2022, doi:10.3390/healthcare10122373_

Round 1

Reviewer 1 Report

First of all, I want to note that it has been a pleasure review your manuscript. I think this is an interesting topic for clinicians who manage the prevalent condition of upper limb musculoskeletal disorders.

The study investigates the prevalence of these disorders and their association with smartphone use in students during the COVID-19 pandemic.  

After reading in depth the manuscript, I would like to make some comments and ask the authors several questions about.

-The large number of authors is striking. This could not be beneficial in terms of accreditation. We recommend that next time the work is carried out with a smaller number of authors.

-Line 43 is incorrectly terminated. Please go through the whole document because there are more incorrectly terminated lines.

- correct the space between COVID-19 and pandemic. Revise entire document...line 121, 191, 234, 330, 331

- correct, please, line 60: …”elbow in (14.1% to 15%) the university students “.

- In the sentence between line 64 and 65 it says "Most of the earlier research aimed to study..." to which articles it refers, should be referenced.

- Line 116:  Some reference should be made when referring to National Institute for Occupational Safety and Health the agency in the text.

- There is no mention in the text of any information or anything that introduces table 5.

-  In the discussion section the idea expressed between the line 224 and the line 226 is not understood because it compares with the study of Australia and Thailand: these articles deal with the same subject?  It is not clear as it stands.

- in line 276: “Earlier researchers are mentioned online but only one reference is made, i.e. only one article is mentioned.

Reviewer 2 Report

It has been a comprehensive study investigating the rapidly increasing phone use in recent years and its effects on health. The work is generally well designed and written. In terms of grammar,  it needs to be reviewed by a native English speaker.

Author Response

Comment: It has been a comprehensive study investigating the rapidly increasing phone use in recent years and its effects on health. The work is generally well designed and written. In terms of grammar,  it needs to be reviewed by a native English speaker.

Response: MDPI Language editing service was utilized to perform the language editing for the entire manuscript.

Reviewer 3 Report

I would like to thank you for the opportunity to review this manuscript. This manuscript aims to know the prevalence of upper limb musculoskeletal disorders and their association with smartphone addiction and smartphone usage among university students.

1.      Study population
How many
students aged between 18-45 years from the departments of the College of Applied Medical Sciences, Majmaah University? The authors should provide information on participant rate, and discuss the selection bias.

2.      Statistical analysis
The authors should describe how did they construct the adjusted model.

3.      Results, Tables 3-8
3.1.  How did the authors estimate the p-value? Chi-square test?
3.2.  Values of OR should not be negative.
3.3.  Smart phone and other gadget usage. There was NO reference.

Relatively minor comments
4.  Title
It may include the study design (cross-sectional study) in the title.

5.  Abstract
The authors should include the study period (year).

Round 2

Reviewer 3 Report

Thank you very much for the response. I have two additional comments.

In Tables 3-8, the authors should present information which factors did they include in the adjusted model.

Several ORs had very wide range of 95% CI. How do the authors interpret these estimates?

Author Response

Comment1: In Tables 3-8, the authors should present information which factors did they include in the adjusted model.

Response: The details were added as a footnote under each table and highlighted in red.

Comment 2: Several ORs had very wide range of 95% CI. How do the authors interpret these estimates?

Response: The interpretation was discussed as a limitation with a reference and highlighted in red.